# A General Use QSAR-ARX Model to Predict the Corrosion Inhibition Efficiency of Drugs in Terms of Quantum Mechanical Descriptors and Experimental Comparison for Lidocaine

**DOI:** 10.3390/ijms23095086

**Published:** 2022-05-03

**Authors:** Carlos Beltran-Perez, Andrés A. A. Serrano, Gilberto Solís-Rosas, Anatolio Martínez-Jiménez, Ricardo Orozco-Cruz, Araceli Espinoza-Vázquez, Alan Miralrio

**Affiliations:** 1Tecnologico de Monterrey, Escuela de Ingeniería y Ciencias, Ave. Eugenio Garza Sada 2501, Monterrey 64849, Mexico; carlos.beltran@tec.mx (C.B.-P.); andres.serrano@tec.mx (A.A.A.S.); gilsolisr@gmail.com (G.S.-R.); 2Departamento de Ciencias Básicas, División de CBI (Ciencias Básicas e Ingeniería), Universidad Autónoma Metropolitana, Unidad Azcapotzalco, Área de Física Atómica Molecular Aplicada, San Pablo 180, Ciudad de México 02200, Mexico; amartinez@azc.uam.mx; 3Unidad Anticorrosión, Instituto de Ingeniería, Universidad Veracruzana, Boca del Río 94292, Mexico; rorozco@uv.mx

**Keywords:** corrosion inhibition, lidocaine, QSAR, tight binding, ARX model, FROLS algorithm, lidocaine

## Abstract

A study of 250 commercial drugs to act as corrosion inhibitors on steel has been developed by applying the quantitative structure-activity relationship (QSAR) paradigm. Hard-soft acid-base (HSAB) descriptors were used to establish a mathematical model to predict the corrosion inhibition efficiency (IE%) of several commercial drugs on steel surfaces. These descriptors were calculated through third-order density-functional tight binding (DFTB) methods. The mathematical modeling was carried out through autoregressive with exogenous inputs (ARX) framework and tested by fivefold cross-validation. Another set of drugs was used as an external validation, obtaining SD, RMSE, and MSE, obtaining 6.76%, 3.89%, 7.03%, and 49.47%, respectively. With a predicted value of IE% = 87.51%, lidocaine was selected to perform a final comparison with experimental results. By the first time, this drug obtained a maximum IE%, determined experimentally by electrochemical impedance spectroscopy measurements at 100 ppm concentration, of about 92.5%, which stands within limits of 1 SD from the predicted ARX model value. From the qualitative perspective, several potential trends have emerged from the estimated values. Among them, macrolides, alkaloids from *Rauwolfia* species, cephalosporin, and rifamycin antibiotics are expected to exhibit high IE% on steel surfaces. Additionally, IE% increases as the energy of HOMO decreases. The highest efficiency is obtained in case of the molecules with the highest *ω* and Δ*N* values. The most efficient drugs are found with p*K*_a_ ranging from 1.70 to 9.46. The drugs recurrently exhibit aromatic rings, carbonyl, and hydroxyl groups with the highest IE% values.

## 1. Introduction 

### 1.1. Corrosion Inhibition and QSAR Fundamentals

Amongst the metals, steel is the most used iron alloy for industrial applications [1], such as oil, food, energy, chemical, and construction industries. Being highly ductile, durable, and resistant, steel is highly appreciated for its mechanical properties. Furthermore, the several different alloys obtained at a considerable low-cost increase the variety of properties exhibited. Unfortunately, corrosion is probably the most common phenomenon that leads to weakening metals; this originates from the electrochemical interaction of metallic surfaces with a corrosive environment. Furthermore, the sulfates, oxides, and other compounds produced from these interactions modify the inherent properties of the metal surface, leading to undesired behaviors [2].

From the above, the most common corrosion inhibition strategies are dedicated to steel [3,4,5], although copper and aluminum alloys are studied to a lesser extent [6,7,8,9,10,11,12]. According to the National Association of Corrosion Engineers (NACE), through its “International Measures of Prevention, Application and Economics of Corrosion Technology” study, the Global cost of the damages produced by corrosion in 2013 was US$2.5 trillion. This massive amount of resources represented 3.4% of the Global Gross Domestic Product (GDP) in 2013 [13]. Thus, providing novel solutions to reduce the undesired effects of corrosion on metals is a global priority.

The most recurrent strategy to reduce the corrosion on metals is to employ corrosion inhibitors on their surface [3,14,15,16]. A corrosion inhibitor is a substance that, added in small amounts to the metal surface, reduces the action of the corrosive media on the metal by forming a protective film. These organic species could bond strongly to the metal surface through intermolecular interactions, from the inhibitor molecule to neighboring metal atoms [2]. Electrostatic interactions, London dispersion forces, and even covalent bonding could be exhibited. Mostly, these organic compounds contain nitrogen, oxygen, and sulfur. Moreover, organic molecules rich in *π*-electrons, associated with either triple, double, or conjugated bonds and aromatic rings, are recurrently found to act as corrosion inhibitors [2,3,4,5,17].

Several organic compounds acting as corrosion inhibitors have been evaluated in recent years—for instance, plant extracts [14,17,18,19,20,21,22,23] and commercial drugs [3,4,5,6,7,8,10,24,25,26,27,28,29,30]. However, the massive amounts of phytochemicals and drugs that can be tested as corrosion inhibitors need intelligent strategies for their study [31,32,33]. In pharmacology, the evaluation and prediction of biological activities and properties for massive amounts of potential drugs have been assessed for a long time by the quantitative structure-activity/property relationship (QSAR/QSPR) paradigm [34,35,36,37]. QSAR/QSPR proposes predicting a given activity/property of quantifiable descriptors. These values can be extracted from existing databases, experiments, theoretical calculations, or simulations. Consequently, data curation, descriptors selection, and mathematical modeling join the problem proposed by the QSAR/QSPR approach [38].

Besides, the QSAR/QSPR modeling in the corrosion field is scarce. The ground-breaking work of Zhang et al. proposes QSAR linear models correlating parameters, obtained mainly by quantum-chemical calculations, such as polarizability, dipole moment, frontier orbital energies, and others, to predict the corrosion inhibition efficiency (IE%) of 18 inhibitors on steel, obtaining average deviations of about 9.82%. Another relevant and quite elegant QSAR linear model is the one reported by Keshavarz and coworkers [39], given the number of nitrogen atoms, amino groups, and other structural parameters. This QSAR model accounts for root mean squared deviation (RMSD), mean absolute error, and maximum errors of about 6.15%, 4.93%, and 12.0%, respectively.

Moreover, recent reports on QSAR/QSPR studies applied to predict IE% of organic corrosion inhibitors on metal surfaces borrow tools from data science. For instance, Liu and coworkers used support vector machine (SVM) models with 11 top descriptors to characterize 20 benzimidazole derivatives [40]. The root mean square error (RMSE) reported is about 4.45%. On the other hand, Ser et al. reported that the corrosion inhibition efficiency of pyridines and quinolines on iron surfaces had been evaluated utilizing machine learning-based QSPR relationships [41]. The authors obtained the mathematical model by genetic algorithm-artificial neural network methods, leading to linear and nonlinear models, with RMSE of about 16.5% and 8.8%, respectively. In this case, the models considered up to nine variables, mainly obtained from DFT calculations.

### 1.2. QSAR Paradigm and HSAB Descriptors

This work proposes modeling the quantitative structure-activity relationship (QSAR) paradigm using several empirical and theoretical descriptors to predict organic molecule corrosion inhibition efficiency. Since these descriptors are almost any quantitative values that can be measured by experiments or determined by theoretical calculations, this work proposes a set of empirical and theoretical descriptors.

The QSAR descriptors determine a drug’s biological activity, looking for specific behaviors. For instance, as proposed by Hansch and Muir, the log *P* octanol-water partition coefficient was correlated with the biological activity 1/*C* [35]. The log *P* descriptor is massively used to study potential drugs since it determines a substance’s concentrations between a hydrophobic phase and a hydrophilic phase. Other QSAR descriptors commonly used in drug design are the solubility coefficient log *S*, p*K*_a_, molecular weight, polar surface area, polarizability, and H-bond acceptor and donor counts [35]. Some common QSAR descriptors are defined within Pearson’s HSAB [42] theory. These descriptors are based on the vertical ionization energy (*I*) and the vertical electron affinity (*A*). By Koopmans’ theorem, both can be calculated by *I* = −*E*_HOMO_ and *A* = −*E*_LUMO_, where *E*_HOMO_ and *E*_LUMO_ are the energies of the highest occupied molecular orbital (HOMO) and the lowest unoccupied molecular orbital (LUMO), respectively. According to the HSAB principle, two species will interact easily if either hard or soft [42]. A molecule’s global hardness (*η*) can be calculated by *η* = (*I* − *A*)/2. Other helpful properties are the absolute electronegativity *χ* = (*I* + *A*)/2 and the global electrophilicity *ω* = *μ*^2/^2*η* [43]. The first one determines the “ability” of a given molecule to attract electrons from the environment, whereas the second is related to the energetic stabilization a species gains by obtaining an additional electron.

Lastly, the fraction of electrons transferred (Δ*N*) is a valuable property to elucidate the behavior of the organic molecule interacting with the metal surface. This value can be calculated by Δ*N* = (*χ*_Metal_ − *χ*_Inhibitor_)/[2(*η*_Metal_ + *η*_Inhibitor_)], where the electronegativities and harnesses of the metal species and the corrosion inhibitor molecule are used. Lukovits and coworkers reported that the higher Δ*N* value, the higher the corrosion inhibition efficiency [44]. Additionally, isosurfaces of frontier molecular orbitals and electrostatic potential arise to describe the active sites and the electrostatic interactions [45].

This work elucidated the contribution to the corrosion inhibition properties in terms of these HSAB descriptors. In addition, the most common QSAR descriptors listed in the previous subsection were studied for completeness. Since DFT calculations require massive computational resources, third-order density-functional tight binding (DFTB) method with Lennard-Jones (LJ) dispersion-correction was used to obtain the quantum chemical descriptors. Thus, the method, hereafter labeled as DFTB3-LJ, was used as implemented in the DFTB+ 21.2 quantum chemistry package [46,47] (more details in Appendix A).

From all the above, this work aims to obtain a QSAR model to predict common commercial drugs to act as high-performance corrosion inhibitors. Thus, this work is divided into four subsections shown as follows. Firstly, the QSAR model is rationalized. Secondly, we describe how the corrosion inhibition efficiency behaves in terms of the descriptors chosen. Then, we discuss the characterization of the high-performance corrosion inhibitors by a quantum-chemical study, and lastly, we provide a theoretical-experimental evaluation of lidocaine as a corrosion inhibitor on steel.

## 2. Theoretical and Experimental Methods

### 2.1. NARMAX System Identification Approach

Nonlinear autoregressive moving averages with exogenous inputs, NARMAX, system identification methodology [48] is used to build models smartly using historical data from the inputs, outputs, and other components. Some relevant applications of these techniques are stock prices [49] and weather [50] prediction, speech recognition [51], pattern classification [52], and aircraft dynamics [53]. As a model, the autoregressive with exogenous inputs (ARX) model is more straightforward (and easier to solve) than other more complex models such as NARMAX. This simplicity is because the ARX model does not require more detailed information about the system to be identified, such as nonlinear elements, which is why it is considered convenient for engineering applications where sufficient data in terms of representativeness is attainable as monitoring or diagnosis [54].

The NARMAX system identification methodology builds first a dictionary or matrix *D* composed of the system descriptors and corresponding observations. At that point, the forward regression orthogonal least squares (FROLS) algorithm [55,56] and the error reduction ratio (ERR) estimator [57] process the *D* matrix to produce an iterative feature selection that is capable of building a compact but representative model.

The ARX model in the system identification context has been successfully applied in various areas, such as troubleshooting and diagnostics for cooling dehumidifiers [54], fault analysis modeling for variable air volume [58], monitoring of buildings’ energy consumption [59], interstitial glucose prediction during human physical activity [60], to predict the global magnetic disturbance in near-Earth space [61], the variability of the Atlantic meridional circulation [62], the Artemia population swimming motion [63], Drosophila photoreceptor responses [64], and EEG signal identification in the neuroscience field [65].

### 2.2. ARX Theoretical Model

The solution method for obtaining the prediction model presented in this work consists of two stages. The first one seeks to place the descriptor data for each drug in the form of a matrix according to the ARX model. In the second part, the descriptors that best explain the IE% are selected in an iterative process to identify the final model.

#### 2.2.1. Arrangement of Candidate Terms

In the first step, it is required to arrange the commercial drug’s data within rows and their corresponding chemical descriptors data into columns, aiming to produce an arrangement analogous to the ARX mathematical model structure, where only linear terms are included. Hence, the drug’s regression representation remains as follows:*y_i_* = Σ*^M^*_*j*__=1_
*β_j_ x_i,j_*∀ *i* ∈ *N*(1)

The previous series can also be expressed as:*y*_*i*_ = *β*_1_
*x*_*i,*__1_ + *β*_2_
*x*_*i,*__2_ + … + *β*_*M*_
*x*_*i,M*_ ∀ *I* ∈ *N*(2)
where *y_i_* is the maximum corrosion inhibition efficiency (IE%) for the *i*th commercial drug contained within the database (for *i* = 1, …, *N*), *x_i,j_* is the *j*th chemical descriptor value for the *i*th commercial drug (for *j* = 1, …, *M*), and *β*_1_, …, *β_M_* is a set of adjusted weights to be computed once the final model has been identified by the FROLS algorithm.

The goal of generating this series of *N* equations is to pack them together to produce an *N ×* 1 vector array on the left side, namely **y**, and an *N × M* matrix arrangement on the right side, namely **X**, containing *N* rows of commercial drugs and *M* columns of chemical descriptors. In this stage, each column in **X** represents a model term that is viewed as a *candidate* that can be included (or not) in the final ARX prediction model, as will be explained in the next section. 

#### 2.2.2. FROLS and ERR Algorithms for Model Structure Selection

The second stage of the solution methodology aims to identify a final ARX prediction model for accurately predicting the corrosion inhibition efficiency. We incorporated the FROLS algorithm to achieve this end, designed to select a subset of candidate terms that best explains the corrosion inhibition efficiencies contained in vector **y**.

In the final model selection, the FROLS algorithm compares each candidate column in **X** with the column vector **y** to incorporate step by step into the model the terms/features that primarily reduce the error prediction of **y**. Such a process continues iteratively until the selected descriptors collectively reach a minimum value for the sum of error reduction ratio (SERR) [48].

Finally, to complete the system’s mathematical model, it is necessary to obtain the weights or parameters of the descriptors previously selected by the FROLS algorithm. The weights are easy to calculate thanks to the fact that the new model has a linear-in-the-parameters representation:**y** = **β Z**(3)
where **y** is the response vector, **Z** is the matrix of selected descriptors, and **β** is the vector of weights obtained from the model. Thus, to compute the vector of linear weights, we only need to clear the vector **β** as follows:**β** = **y Z**^−1^(4)

#### 2.2.3. Cross-Validation

Cross-validation is a standard procedure when the dataset on hand has too few observations for data splitting. In the case of the current study, fivefold cross-validation was employed to seek a reliable model creation; hereafter, such a method is described [66,67]. Firstly, a database of 250 chemical compounds is elaborated as described in Section 3. Secondly, all compounds containing an experimental value of IE% are promoted to the training/validation set. The remaining compounds, those with no available experimental IE%, are set aside. Then, for each ARX/FROLS simulation run, the training/validation set is further randomly divided into five groups. In the first iteration, the first group is treated as a validation subset, and the remaining four groups are designated as the training subset, over which the ARX model and FROLS algorithm are run. The mean squared error (MSE) is computed on the first validation set. This procedure is repeated five times; each time, a different group of observations is treated as a validation subset. This process results in 5 different linear models with five distinct MSE estimates of the test error. The model with the lowest MSE value is promoted to predict the IE% of the remaining compounds.

Other metrics were employed to analyze the model’s performance: Mean absolute percentage error (MAPE), standard deviation (SD), mean square error, and root mean square error (RMSE). These metrics, discussed throughout the manuscript, are defined now: MAPE = (1/*n*) Σ*^n^_i_*_=1_|(*y_i_* − *ŷ_i_*)/*y_i_*| × 100%, SD = [(1/*n* − 1)Σ*^n^_i_*_=1_ (*ŷ* − *y_i_*)^2^]^1/2^, MSE = (1/*n*) Σ*^n^_i_*_=1_ (*y_i_* − *ŷ_i_*)^2^, and RMSE = [(1/*n*) Σ*^n^_i_*_=1_ (*y_i_* − *ŷ_i_*)^2^]^1/2^. Where *y_i_* is the experimental value of IE% for compound *i*, *ŷ_i_* is the estimated value of IE% for compound *i* provided by the model, and *y* is the average IE% within a sample of compounds of size *n*.

### 2.3. Experimental Details

#### 2.3.1. Solution Preparation

Different concentrations of the lidocaine compound (Figure 1) were prepared and obtained in an injectable solution from “Farmacias del Ahorro” pharmacy. The initial solution concentration was 0.01 M, dissolved in ethanol, to make later solutions of 0, 10, 20, 50, and 100 parts per million (ppm). The corrosive solution is NaCl 3% (100 mL).

#### 2.3.2. Electrochemical Evaluation

The standard three-electrode system was used for the electrochemical evaluation at room temperature. The API 5L X70 sample was the working electrode, a saturated Ag/AgCl electrode was the reference electrode, and a graphite bar was the counter electrode. The test sequence was performed on a piece of Gill-AC equipment as follows: (a) Open circuit potential (OCP) was measured for 1800 s; (b) Electrochemical impedance spectroscopy (EIS) was employed using 10^−2^–10^4^ Hz with an amplitude of ±10 mV. The exposure area of experimentally used samples was 0.78 cm^2^. The electrochemical tests were performed in triplicate.

After the EIS measurements, potentiodynamic polarization curves of the inhibitor at different concentrations were performed, measured from −300 mV to 300 mV in relation to the open circuit potential (OCP), with the speed of 60 mV/min using the ACM Analysis software for data interpretation.

#### 2.3.3. Characterization by Atomic Force Microscopy (AFM)

The morphology of the steel sample surface after immersion in the corrosive media, in the presence and absence of the Lidocaine inhibitor, was characterized by atomic force microscopy (AFM) using digital instruments scanning probe microscope with a nanoscope IIIa controller. The AFM was operated in tapping mode using an etched silicon cantilever with a length of 125 µm, with a nominal tip radius of approximately 10 nm.

## 3. Results and Discussion

In order to predict the corrosion inhibition efficiency of drugs on steel surfaces, a database with 250 commercial drugs containing common QSAR descriptors and those formulated within Pearson’s HSAB theory was used to obtain a linear mathematical model by an ARX analysis. The ARX methodology was also used to exclude the variables that do not contribute to the prediction of IE%. In addition, a more sophisticated method for data analysis, IBM’s Watson artificial intelligence [68], will be implemented to compare it with the ARX model and ensure the proper performance of the linear function. Thus, this work is divided into four subsections. Firstly, the model’s determination is discussed and compared briefly with the privative AI model. Secondly, the main tendencies exhibited by the linear model are discussed. Then, species predicted as highly efficient corrosion inhibitors are studied and, for extension, correlated with their families. Lastly, the prediction of IE% for lidocaine was compared with its experimental counterpart.

### 3.1. Model Determination

This section explains how an ARX model was estimated to predict the iron corrosion inhibition of different compounds through a system identification methodology based on the FROLS algorithm. First, we narrate the data processing of the stated problem into a linear input-output system, followed by a term selection process to reach the final prediction model.

#### 3.1.1. Data Processing into an ARX Linear System

As stated, system identification aims to find a model that reveals the distinctive elements of a system by processing the history of its interactions with the environment. Here, we took the system to be identified as the corrosion inhibition efficiency on steel, IE%, explained by ten candidate quantum chemical descriptors. We studied the previous interaction through instances collected from 42 commercial chemical substances. The input-output model determination problem implies a suitable selection of the variables (descriptors) present in the following simple ARX linear model:*y* = *β*_1_
*x*_1_ + *β*_2_
*x*_2_ + *β*_3_
*x*_3_ + *β*_4_
*x*_4_ + *β*_5_
*x*_5_ + *β*_6_
*x*_6_
*+ β*_7_
*x*_7_ + *β*_8_
*x*_8_ + *β*_9_
*x*_9_
*+ β*_10_
*x*_10_(5)
where *y* is an output term plus ten input terms composed of candidate descriptors, as labeled in Table 1, and a corresponding parameter value or weight *β_i_*. The list of the ten candidate descriptors and their corresponding index number is listed in Table 1. Each drug’s molecular weight (MW) was considered a size-dependent parameter. Additionally, the negative base-10 logarithm of the acid dissociation constant of a solution, p*K*_a_ = −log*K*_a_, was included to determine the strength of an acid in the solution. Additionally, the octanol-water partition coefficient, log *P*, is a descriptor associated with the concentration of a given substance in the aqueous phase of a two-phase octanol-water mix. Similarly, the log *S* descriptor is directly related to the water solubility of a substance employing a base-10 logarithm. Besides, the polar surface area (PSA) is the molecular surface associated with heteroatoms and polar hydrogen atoms, giving a quantitative amount related to charge accumulation. In addition, polarizability, α, denotes the tendency of a particular molecule to acquire an electric dipole moment in the presence of an external electric field. As described previously, energies of HOMO and LUMO orbitals can be related, through Koopman’s theorem, to ionization energy and the electron affinity of a given molecular species, respectively. In addition, electrophilicity, *ω*, relates to the change in energy of an electrophile when it comes in contact with a perfect nucleophile, being a measure of the tendency to react between electrophile and nucleophile species. Finally, the fraction of electrons shared, Δ*N*, was selected since it relates to the amount of charge transferred from one species to another (Table 1).

Unlike nonlinear models with second or third-order expansions, linear regression models are simpler to solve as they have precisely the same number of variables and model terms. A final critical point in data processing is the determination of the partition of the samples in testing and training data sets. Here, we divided data at random, where 80% of the data was used for training and 20% for validation.

#### 3.1.2. Term Selection through FROLS and ERR

In the term selection stage, we consider the linear model in Equation (5) and the training set as entry points to the FROLS algorithm, which can easily detect the most relevant terms in first-order expansions [48]. At the first step, the ERR values of each candidate m included in the *D* dictionary, where *D* = {**p**_1_, **p**_2_, …, **p***_M_*}, are determined:ERR*_m_ =* (**y**^T^**p**_m_/**p**^T^*_m_***p***_m_*)^2^(**p**^T^*_m_***p***_m_*)/**y**^T^**y**(6)

The ERR value helps determine the final subset of the model’s terms because these indicate each one’s contribution to the prediction error reduction. The first term selected from the *D* dictionary is always the one with the highest ERR. The descriptor with the highest ERR was *x*_7_, with an ERR of 97.82% in our testing.

The selection process takes only the remaining unselected terms from dictionary *D* and introduces orthogonal transformations via the Gram–Schmidt algorithm from step two onwards. The orthogonalization process prevents the following candidates from being included from containing information already provided by the descriptors already selected, thus each new term contributes independently to the model’s accuracy. Finally, the selection procedure ends when the error-to-signal ratio (ESR) decreases below a predetermined threshold, where:ESR = 1 − Σ*^M^*^o^*_i_*_=1_ ERR*_i_* ≤ *ρ*(7)
where *ρ* is a very small value, for instance, in this work, *ρ* = 0.005 and *M*_0_ is the number of unselected candidate variables. The terms included in the final model and the ERR values for each can be consulted in Table 1.

The final calculation involved the estimation of the parameters **β**, as it is shown in Section 2.2.1, thus the final ARX model, derived from the initial linear formulation in (5), can be stated as follows:*ŷ* = 812.1748 *x*_7_ + 33.1669 *x*_10_ + 823.4630 *x*_8_ + 6579.0080 *x*_9_ + 0.5287 *x*_2_(8)

Consequently, the linear ARX model obtained accurate and precise results compared to the testing set, as evidenced by the computed mean absolute percentage error and standard deviation. The testing set included aspirin, cephapirin, ascorbic acid, imidazole, trimethoprim, clindamycin, phenobarbital, and doxycycline data, obtaining MAPE, SD, RMSE, and MSE of about 5.18, 2.51, 4.87, and 23.80%, respectively.

In addition, another set of drugs was collected to verify the generalization power of the ARX model. Additional data for streptomycin [73], fexofenadine [74], quinoline [75] N, N-dimethylformamide [76], and mycophenolic acid [25] was used to compute the MAPE, SD, RMSE, and MSE, obtaining 6.76%, 3.89%, 7.03%, and 49.47%, respectively. These values are fully comparable to those obtained with the testing set, pointing to a correct prediction of IE% values for drugs relatively different from those in the original database. IBM’s Watson artificial intelligence platform was also used to obtain a private and highly hyperparametrized model. In this case, the five variables (p*K*_a_, *E*_HOMO_, *E*_LUMO_, *ω*, and Δ*N*) in the ARX model were included in the AutoAI Watson’s routine to fit the experimental IE% values [77]. Additionally, 80% of the data was used for the training set and the remaining 20% for validation. Finally, four different experiments, pipelines, were done by the extra trees regressor algorithm. The model obtained improved the external comparison only to 5.44%, 2.91%, 5.35%, 28.59% for MAPE, SD, RMSE, and MSE, respectively. Thus, the ARX model (obtained through FROLS), a linear function that depends only on five descriptors, obtained results close to a highly hyperparametrized, non-portable, and nonlinear alternative.

Other QSAR models have been proposed recently. Thus, it is pertinent to compare our linear ARX model with those found in the literature regarding standard evaluation metrics. For instance, Quadri and coworkers reported several multiple linear regression and artificial neural network (ANN) models adjusted to twenty pyridazine derivatives. The best ANN model yielded a lower MAPE value, of about 10.2362%, whereas the RMSE and MSE achieved were 10.5637 and 111.5910%, respectively [78]. In comparison, Li and colleagues obtained a QSAR model by a support vector machine approach to predict the performance of benzimidazole derivatives. In this case, the nonlinear model achieved an RMSE of about 6.79% [79]. More recently, they updated the model, improving the RMSE up to 4.45% [40]. In addition, Al-Fakih and coworkers reported QSAR models for furan derivatives, obtained with sparse multiple linear regression using ridge penalty and sparse multiple linear regression using an elastic net, achieving MSE of about 7.75 and 2.34%, respectively [80]. The ARX approach obtained similar metrics compared to those obtained with alternative methods, as reported by other authors.

A brief discussion is included below to identify the extent and perspectives of the ARX linear model. In principle, corrosion inhibition is a multifactorial phenomenon since drug solubility, pH, temperature, concentration, corrosive medium, dynamic conditions, the employed alloy, and even experimental technique used to determine IE% could influence the results [2]. Thus, it is possible to assume that a 5-parameters model such as that introduced above is not enough to catch the variety of conditions experimentally used. However, to the best of our knowledge, these experimental variables are not recurrently considered, possibly by the scare IE% values measured with the same experimental design, hindering the formulation of mathematical models. Nevertheless, work conditions are naturally occurring variables that should be considered for robust predictive models.

Finally, although linear models to predict IE% are the most common approach [78], nonlinear formulations can also be suitable. A nonlinear version of ARX comes to be the NARMAX model in the current case. According to Gu et al., nonlinear models performed better than linear ones in a study about cortical responses. Nevertheless, the linear terms had larger weights than those in the resulting NARMAX models [81]. In addition, NARMAX approaches are known to identify mathematical models for nonlinear systems, which prevail in nature. This is the case of the solar wind coupling analysis reported by Boynton and coworkers [82] or the peak air pollution levels forecasted by Pisoni et al. [83]. Thus, it is expected that a NARMAX approach to the corrosion inhibition problem may well determine whether the phenomenon is nonlinear in terms of the proposed descriptors. However, as suggested by Boynton, a high number of possible monomials resulting from the polynomial expansion can be a challenging situation. The above stems from the need for a final parsimonious NARMAX model with fewer monomials selected out of a vast majority with no or minimal influence on the phenomenon. In principle, such as in the ARX approach used here, the FROLS algorithm could lead to a small set of monomials within the selected allowable model order [84].

### 3.2. Main Tendencies

This subsection aims to elucidate the main tendencies exhibited by the predictions, carried out on 250 commercial drugs by the ARX model detailed previously, of the corrosion inhibition efficiency IE%. Thus, five variables contained in the ARX mathematical model (*E*_HOMO_, *E*_LUMO_, Δ*N*, *ω*, and p*K*_a_) and molecular weight were used to illustrate the predicted IE% values. In addition, five drugs were excluded from the following analyses due to their unrealistic IE% values predicted above 100%. These species are sulfadiazine (106.31%), methacycline (111.72%), glycine (124.03%), ethosuximide (158.88%), and hexetidine (259.25%).

First of all, Figure 2 shows that corrosion inhibition efficiency IE% increases as the energy of HOMO decreases. Consequently, by Koopman’s theorem, the most efficient drug molecules to act as corrosion inhibitors are those with the lowest ionization potential. The above can be rationalized by the necessity of the metal surface atoms to fill their vacant d-orbitals with electrons, coming from the corrosion inhibition molecule in the current case. Consequently, the easier it is to remove valence electrons from the organic molecule, the higher corrosion inhibition performance exhibited.

On the other hand, *E*_LUMO_ splits the IE% values into two sets (Figure 2). The first set of drugs, with *E*_LUMO_ values below 2.0 eV, shows a tendency similar to that exhibited by *E*_HOMO_ since the most efficient corrosion inhibitors are those with the most negative *E*_LUMO_ values and consequently the highest electron affinities. Thus, it is expected that high-performance corrosion inhibitor molecules can catch electrons from the environment and donate them, leading to shared electrons as in covalent interactions. However, the other set of molecules with moderate performance, with *E*_LUMO_ values above 4.5 eV, offers another route to produce the corrosion inhibition effect (Figure 3). Since up to 14 species are obtained with low and even positive *E*_LUMO_ values and intermediate *E*_HOMO_ energies, ranging from −6.5 to −5.0 eV, they are expected to donate charge to the metal surface and handle mostly electrostatic interactions with it. These intermediate efficiency corrosion inhibitors are, in order of increasing IE%, the following ones: cyclopentamine (84.92%), methenamine (85.56%), triethylamine (86.74%), gentamicin (87.12%), mecamylamine (88.32%), kanamycin (88.66%), diethylamine (90.00%), diethanolamine (90.75%), ethambutol (91.46%), amantadine (92.43%), ethanolamine (92.57%), tromethamine (92.59%), tuaminoheptane (92.99%), and ethylamine (94.12%).

In the case of IE% as a function of the electrophilicity and the fraction of electrons shared, the efficiency shows two sets of compounds. The highest IE% values are obtained in the molecules with the highest *ω* and Δ*N* values (see Figure 3). The above can be related to the previous assumptions about how the electrons behave between the corrosion inhibitor molecule and the metal surface. Thus, a highly effective corrosion inhibitor molecule is expected to donate a considerable number of electrons to the metal surface, as denoted by the Δ*N* values calculated for the specific case of an iron surface. Additionally, according to the ARX model, a highly efficient corrosion inhibitor molecule is expected to behave as an electrophile with high power to attract electrons to itself. Besides, the other set of molecules is composed precisely of those drugs named above. Drugs with intermediate performance, with IE% efficiency ranging from 84.92% to 94.12%, cannot donate large amounts of charge to the metal surface and neither to attract it. Thus, the fourteen species with moderate performance are expected to interact by electrostatic interactions.

Lastly, the heat map obtained for IE% as a function of the molecular weight and p*K*_a_ shows more smoothly dispersed values without the two sets obtained previously (see Figure 4). It is noticeable that the highest efficiency is obtained for species with molecular weight ranging from 415 to 823 Da. Additionally, all the high-performance molecules are predicted for those species with positive p*K*_a_, ranging from 5 to 10. Thus, weak acids are expected to behave as potential corrosion inhibitors, whereas strong acids are not helpful for this application. Although light molecules are not particularly prominent by their corrosion inhibition efficiencies, values above 90% are easily reached. In the case of these light species, with a molecular weight below 400 Da, the IE% increases as the p*K*_a_ increases. Since the commercial drugs must exhibit high corrosion inhibition efficiencies, above 90% according to international standards for industrial applications, the following subsection aims to deep into the species predicted by the ARX models as highly efficient corrosion inhibitors.

### 3.3. High-Efficiency Corrosion Inhibitors

In order to produce reliable predictions for highly efficient commercial drugs to act as corrosion inhibitors, species with IE% estimated above 95% are studied as follows. The value 95% was chosen to take into account MAPE and SD values computed for the mathematical model of about 5.18% and 2.51%, respectively. This way, it is expected that efficiencies measured under experimental conditions must fall into the regime required for industrial applications, with inhibition efficiencies above 90%. The drugs fulfilling these conditions, shown in Table 2, are minocycline (97.58%), deserpidine (95.29%), daunorubicin (96.67%), dipyridamole (97.28%), doxorubicin (97.40%), amphotericin B (97.55%), acepromazine (97.73%), cephaloridine (98.57%), mercaptopurine (98.66%), and rifampicin (98.71%).

Being weak acids, the most efficient drugs are found with p*K*_a_ ranging from 1.70 to 9.46. Moreover, *E*_HOMO_ values are found in a tight range, from −5.87 to −4.34 eV. Conversely, *E*_LUMO_ values are exhibited in a broader range, from −4.01 to −1.83 eV. Electrophilicity ranges from 0.77 to 1.23 eV, whereas Δ*N* ranges from 1.12 to 1.55. Thus, all the above is consistent with previous observations about general tendencies. There are no significant similarities among all the high-efficient species; however, some functional groups are recurrently exhibited by these species. In addition, invoking the similarity principle, drugs belonging to the same family are suitable to be presumed as corrosion inhibitors with efficiencies comparable to those obtained for the species shown in Table 2. For instance, minocycline is a tetracycline antibiotic, with multiple dimethylamino, hydroxyl, and carbonyl groups, obtained as a highly efficient corrosion inhibitor, with an IE% predicted value of 97.58% (Table 2). Other tetracycline antibiotics are presumed to be obtained as suitable corrosion inhibitors. Doxycycline and oxytetracycline are predicted as efficient CIs, with 91.89% and 93.95% IE% values, respectively.

Deserpidine is an example of an ester alkaloid drug, exhibiting multiple methoxyl and carbonyl groups, used for their antihypertensive and antipsychotic properties. Although alkaloids predicted to act as corrosion inhibitors cover a wide range of structural motifs, the most prominent by their structural similarity to deserpidine and predicted IE% value of about 94.55% is reserpine, another alkaloid that can be obtained from Rauwolfia species (Table 2). By extension, yohimbine is another alkaloid presumed to act as a suitable corrosion inhibitor. Finally, daunorubicin and doxorubicin are two examples of anthracycline class drugs for cancer treatments [85], with predicted IE% values of about 96.67% and 97.40%. Similar to the previous cases, these species exhibit several hydroxyl and carbonyl groups, presumed to be covalent-polar and dative bonds with iron surface atoms, respectively [86,87]. Additionally, multiple π-electrons coming from their aromatic rings can be donated to the iron surface.

Furthermore, amphotericin B is another drug that is predicted to act as a highly efficient corrosion inhibitor, with IE% estimated as 97.55%. This drug belongs to macrolides and the same class of antibiotics, including erythromycin, roxithromycin, azithromycin, and clarithromycin. In particular, Amphotericin B does not contain several aromatic rings, such as those exhibited by the species previously discussed. Thus, this drug, exhibiting eleven hydroxyl groups and two carbonyl ones, is expected to interact mainly with the metal surface by their functional groups. Erythromycin is another macrolide expected to work as a suitable corrosion inhibitor, with IE% estimated as 90.11% (Table 2) according to NRF-005-2009. This decrement in the corrosion inhibition efficiency can be rationalized by the few functional groups in erythromycin capable of interacting with the metal surface compared to amphotericin B.

On the other hand, acepromazine heads the phenothiazine family of antipsychotics, accounting for an estimated corrosion inhibition efficiency of about 97.73%. This molecule’s interactions can be explained by the π electrons available in its aromatic rings in addition to a carbonyl group. Other members of the phenothiazine family, promazine and levomepromazine, lack that carbonyl group. These drugs achieve lower corrosion inhibition efficiencies of 91.19 and 91.81% (Table 2). Another relevant drug is cefaloridine, being part of the cephalosporin class, a large group of antibiotics derived from the fungus acremonium. Aromatic rings and carbonyl groups are the most relevant structural motifs constituting cefaloridine. However, several cephalosporin drugs were also studied but achieved lower efficiencies. These drugs are cephapirin (85.72%), cephalexin (85.87%), cephalothin (86.06%), cefazolin (86.24%), and cephradine (86.68%). In this case, it is unclear why cefaloridine is so efficient since all cephalosporin exhibits similar functional groups. Thus, it is necessary to deepen the corrosion inhibition effect of cefaloridine and other cephalosporin drugs. According to these measurements, cephapirin (82.5%) and cephalexin (76.9%) achieve lower corrosion inhibition efficiencies in comparison with the predicted values, whereas cephalothin (92.0%), cefazolin (93.9%), and cephradine (95%) are closer to the value predicted for cephaloridine of about 98.57% (see Appendix A). It is plausible that the ARX model does not well describe the cephalosporin family.

Dipyridamole and mercaptopurine are two drugs without relatives in the current study. With high corrosion inhibition efficiencies of about 97.28 and 98.66%, respectively, dipyridamole and mercaptopurine are presumed to strongly interact by their double bonds and polar functional groups. Finally, rifampicin is part of the rifamycins class, a group of antibiotics, with IE% estimated as 98.71% (Table 2). Interestingly, rifampicin is the only of these highly efficient corrosion inhibitor molecules, with an experimental measure of its IE% of about 94.7% (see Appendix A). Furthermore, this species exhibits several functional groups, such as carbonyl and hydroxyl groups and heteroatoms and π-electrons, thus suggesting that rifampicin could interact with the metal surface by covalent, dative bonds, or electrostatic interactions. Thus, the whole rifamycin family could be evaluated as corrosion inhibitors.

The above discussion points out the highly efficient corrosion inhibition properties expected for ten commercial drugs, with IE% values ranging from 95.29 to 98.71% (Table 2). In comparison, other drugs experimentally evaluated exhibited similar or lower corrosion inhibition efficiencies. For instance, losartan, a drug commonly used for hypertension treatment, obtained a maximum IE%, by EIS technique, of only 92.0% [88]. Additionally, salbutamol, a commonly used treatment for asthma, was obtained with a maximum IE%, determined by EIS measurement, of about 84% [89]. Similarly, Anadebe and coworkers collected the maximum IE% achieved by several recently reported commercial drugs, ranging from 80% to 95% [89]. Drugs applied to carbon steel were tobramycin (80%) [90], metformin (90%) [91], metolazone (92%) [92], and nifedipine (94%) [93]. Additionally, species applied on mild steel surfaces were: dexamethasone (83%) [94], rosuvastatin (90%) [95], ambroxol (94%) [96], and dapsone (95%) [97]. Additionally, Abeng, et al. reported the IE% on carbon steel achieved by moxifloxacin (88.2%), nifedipine (89.6%), metolazone (92.8%), and levofloxacin (94.1%) [98]. Even more, the recently evaluated drugs, mycophenolic acid [25] and fluconazole [24], obtained moderate efficiencies of about 90%. Clearly, all these recently studied drugs obtained values below the threshold assumed in this work. However, other drugs were comparable to those expected for the molecules analyzed in this subsection. This is the case of pyrazinamide, isoniazid, and rifampicin, which achieved maximum IE% values of about 95.86%, 97.89%, and 97.06%, respectively [99]. Thus, the predictions and tendencies discussed could be the object of study and further confirmation.

### 3.4. Experimental Verification

(a)Open circuit potential (OCP)

The variations of the OCP without and with the corrosion inhibitor lidocaine reached a steady state at 600 s (Figure 5). It is evident that when a corrosion inhibitor is present, the potential considerably decreases, indicating a drop in the corrosion process.

(b)Concentration effect of lidocaine by EIS

The equivalent electric circuits employed to fit the experimental data of the Nyquist diagrams are shown in Figure 6. A Randles circuit in the case of the sample of Figure 6a was used without inhibitor (Blank). A parallel circuit with two constant phase elements (Figure 6b) for the samples with inhibitor.

Where *R*_s_ is the solution resistance, *R*_ct_ is the charge transfer resistance, *CPE*_inh_ is the constant phase element of the inhibitor, and *CPE*_Rct_ is the constant phase element associated with the double layer. The inhibitor efficiency IE% was calculated by the equation [100]:IE% = 100 [(*R*_p blank_^−1^ − *R*_p inh_^−1^)/*R*_p blank_^−1^)](9)
where *R*_p blank_^−1^ is the polarization resistance of blank and *R*_p in_^−1^ is the polarization resistance of sample with inhibitor.

The polarization resistance (*R*_p_) was calculated with:*R*_p_ = *R*_ct_ + *R*_F_(10)
where *R*_ct_ is charge transference resistance and *R*_F_ film resistance, in Ω cm^2^.

The electrochemical double-layer capacitance (*C*_dl_) was calculated through the next equation [101]:*C*_dl_ = *Y*_0_^1/*n*^ (*R*_s_^−^^1^ + *R*_ct_^−^^1^) ^(*n*^
^−^
^1)/*n*^(11)
where *Y*_0_ is the constant phase element, *R*_s_ is the solution resistance (Ω cm^2^), and *R*_ct_ is the charge transfer resistance (Ω cm^2^).

For the description of a frequency-independent phase shift between an applied AC potential and its current response, a constant phase element (*CPE*) is used, defined in the impedance representation as:Z_CPE_ = *Y*_0_^−^^1^(*j*ω)^−^*^n^*(12)
where *Y*_0_ is the *CPE* constant, *n* is the *CPE* exponent that can be used as a gauge of the heterogeneity or roughness of the surface, *j* = −1 is an imaginary number, and ω is the angular frequency in rad s^−1^. Depending on *n*, *CPE* can represent a resistance (*Z*_CPE_ = *R*, *n* = 0), a capacitance (*Z*_CPE_ = *C*, *n* = 1), and a Warburg impedance (*Z*_CPE_ = *W*, *n* = 0.5), or inductance (*Z*_CPE_ = *L*, *n* = −1). The correct equation to convert *Y*_0_ into *C*_F_ is given by [102]
*C*_F_ = *Y*_0_ (ω′)^n−1^(13)
where *C*_F_ is the film capacitance (µF/cm^2^) and ω′ is the angular frequency at which *Z*_real_ is maximum.

Figure 7a shows the Nyquist diagram of API 5L X52, called “blank”. A semicircle that is not entirely close was observed, attributed to charge transfer resistance.

On the other hand, the Nyquist diagram in Figure 7b in all concentrations showed two processes: one attributed to inhibitor film and the charge transfer resistance [103].

Table 3 shows the electrochemical parameters obtained after fitting with the equivalent electric circuits of Figure 6. At 100 ppm, the highest value of *R*_ct_ was reached (1522 Ω cm^2^). This behavior could be attributed to the high adsorption process of the organic compound on the metallic surface. On the other hand, the *C*_dl_ and *C*_F_ values decreased because the surface is protected by the inhibitor at the metal/solution interface [104].

(c)Polarization curves

The polarization curves of the API 5L X70 steel, immerse in 3% NaCl with and without lidocaine, are shown in Figure 8. The polarization parameters are enlisted in Table 4: corrosion potential (*E*_corr_), current density (*i*_corr_), tafel slopes (b_a_ and b_c_), and the inhibition efficiency (IE%) was determined by:IE% = [1 − *i*_corr_/*i*_corr blank_] × 100(14)
where *i*_corr_ and *i*_corr blank_ are the current density with and without inhibitor.

Figure 8 shows that the curve shifts to the left due to the corrosion current density decreasing when the concentration increases due to the adsorption of the inhibitor.

The electrochemical parameters by this technique show that the corrosion current density (*i*_corr_) value decreased in the presence of the lidocaine inhibitor, being attributed to the protected metal surface (Table 4). This phenomenon implies that the inhibitor might suppress the anodic reaction of the metal dissolution and the detachment of cathodic hydrogen [105], while the inhibition efficiency by this technique shows similar results to the other technique (EIS), and the best concentration was 20 ppm with 92.6% to protect the metal surface. Finally, according to corrosion potential (*E*_corr_), at a concentration of 20 ppm, the lidocaine behaved as an anodic inhibitor, while at 10, 50, and 100 ppm, the behavior was cathodic.

(d)Adsorption process

The corrosion inhibition displaces the water molecules and replaces them with the inhibitor molecules on the metal surface.

Nevertheless, the superficial coverage (*θ*) for the different lidocaine concentrations as corrosion inhibitors in this system was evaluated by EIS using IE%:*θ* = (1/100) IE%(15)

Using the Langmuir isotherm, a good fit is obtained, and according to the value of the free energy of adsorption of Gibbs (Equation (17)), the combined process continues [106].
*C/**θ* = *k*_ads_^−^^1^ + *C*(16)
Δ*G*^0^_ads_ = − 55.5 RT ln *k*_ads_(17)
where *C* is the concentration, *θ* is the coating coverage, *k*_ads_ is the adsorption constant, *R* is the ideal gas constant, and *T* is the temperature. In some reports, the Δ*G*^0^_ads_ will indicate which adsorption mechanism follows the organic compound; if it is lower than −20 kJ/mol, it can be considered a physisorption process. If it is higher than −40 kJ/mol (Δ*G*^0^_ads_ > −40 kJ/mol), then it is a chemisorption process, but if it is in the middle of −20 kJ/mol and −40 kJ/mol, the type of process that is taking place is called “combined”.

Figure 9 shows a good fit, having a correlation coefficient (*R*^2^) of 0.9996. The equation obtained is *C*/*θ* = 1.0534 *C* + 8 × 10^−6^, obtaining a Δ*G*^0^_ads_ = −38.39 kJ/mol due to a combined type process.

(e)AFM analysis

Figure 10 shows AFM images recorded on the surfaces of steel samples, and Table 5 shows the roughness values; both Ra (the mean roughness or arithmetic average of the absolute values of the roughness profile ordinates) and *R*_q_ (root mean square roughness or the root mean square average of the roughness profile ordinates) are reported.

After exposure to the corrosive media for 24 h in the absence (Figure 10a) and in the presence of 50 ppm of lidocaine (Figure 10b), it can be noted that the roughness values of the steel sample that was not protected with lidocaine are notorious compared with that of the sample that was protected with lidocaine or that which was not exposed to the corrosive media, Figure 10c, such notable rugosity is due to the different corrosion products formed in each case.

## 4. Conclusions

A QSAR relationship constructed through a linear ARX model was used to predict the corrosion inhibition efficiency of 250 commercial drugs on steel. The ARX treatment found the five most important descriptors to predict IE%, reducing by half the number of variables used in the linear model. These variables, obtained mostly from quantum chemical calculations of gas-phase molecules at the DFTB3-LJ level, are the p*K*_a_, electrophilicity, HOMO and LUMO energies, and the fraction of electrons transferred to bulk iron.

The ARX model obtained a MAPE and SD of about 5.18 and 2.51%, respectively, compared to the testing set. Another five drugs not included in the original database were used as an external validation set for which the computed MAPE and SD were approximately 6.76% and 3.89%, respectively, thus confirming the previous predictions. In addition, a fivefold model, obtained by IBM’s Watson AI extra trees regressor algorithm, was used to compare it with the ARX one. In this case, IBM’s Watson model improved the external comparison only to 5.44% and 2.91% for MAPE and SD, respectively. Thus, the linear ARX model is competitive compared to highly hyperparametrized and privative alternatives.

Overall, there are several tendencies of IE% as a function of the selected variables. For instance, IE% increases as the energy of HOMO decreases. Additionally, the highest IE% values are obtained in the case of the molecules with the highest *ω* and Δ*N* values. The most efficient drugs are found with p*K*_a_ ranging from 1.70 to 9.46. The drugs recurrently exhibit aromatic rings, carbonyl, and hydroxyl groups with the highest IE% values. Ten drugs are predicted with IE% above 95%—those are: minocycline (97.58%), deserpidine (95.29%), daunorubicin (96.67%), dipyridamole (97.28%), doxorubicin (97.40%), amphotericin B (97.55%), acepromazine (97.73%), cephaloridine (98.57%), mercaptopurine (98.66%), and rifampicin (98.71%). Alkaloids from Rauwolfia species, macrolides, cephalosporin, and rifamycin antibiotics are expected to exhibit high IE% on steel surfaces.

Lastly, lidocaine was predicted and experimentally tested for the first time. At 100 ppm concentration, the lidocaine showed IE% of about 92.5% using electrochemical impedance spectroscopy and 87.4% by polarization curves; in comparison, the ARX model predicted 87.51%. The thermodynamic analysis showed that the lidocaine follows a mixed adsorption process in the API 5L X70 surface. This linear model proposed to deepen the use of commercial and reused drugs to act as corrosion inhibitors on steel.

## Figures and Tables

**Figure 1 ijms-23-05086-f001:**
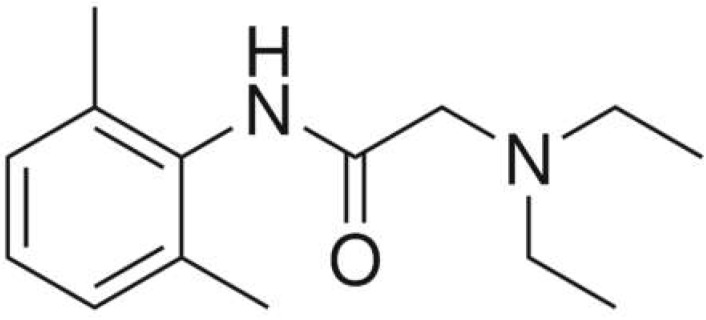
Lidocaine molecule experimentally used to validate the ARX model.

**Figure 2 ijms-23-05086-f002:**
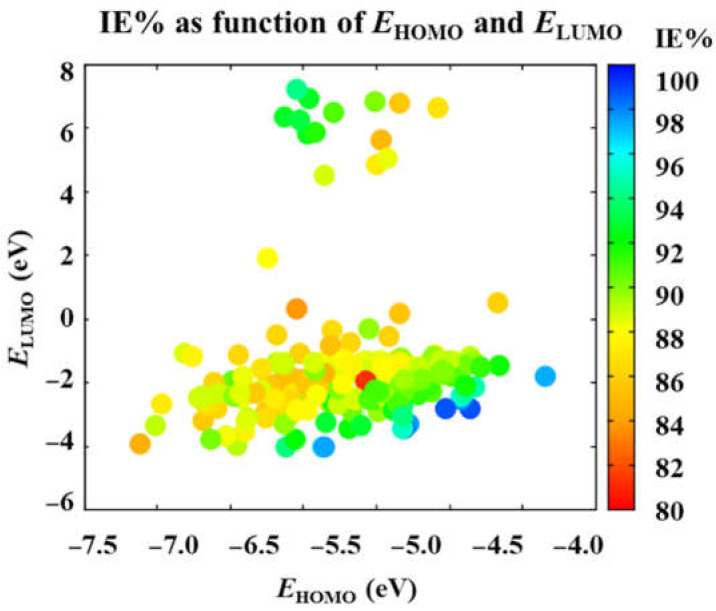
Corrosion inhibition efficiency, predicted by the ARX model, as a function of *E*_HOMO_ and *E*_LUMO_. Variables calculated at the DFTB3-LJ level of theory.

**Figure 3 ijms-23-05086-f003:**
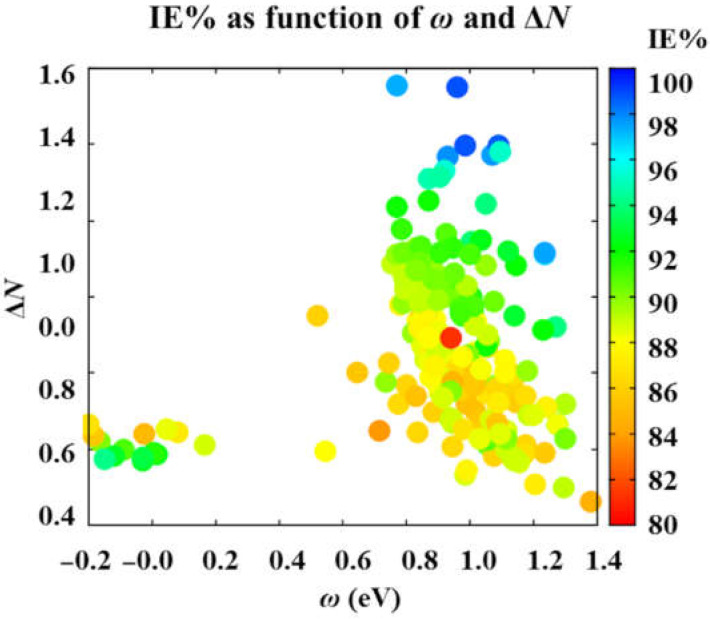
Corrosion inhibition efficiency, predicted by the ARX model, as a function of electrophilicity, *ω*, the fraction of electrons transferred, Δ*N*. Variables calculated at the DFTB3-LJ level of theory.

**Figure 4 ijms-23-05086-f004:**
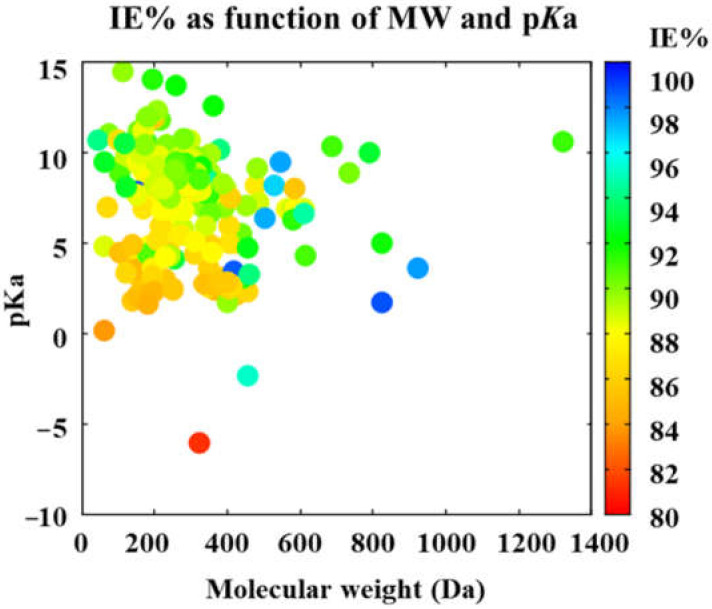
Corrosion inhibition efficiency, predicted by the ARX model, as a function of molecular weight and p*K*_a_.

**Figure 5 ijms-23-05086-f005:**
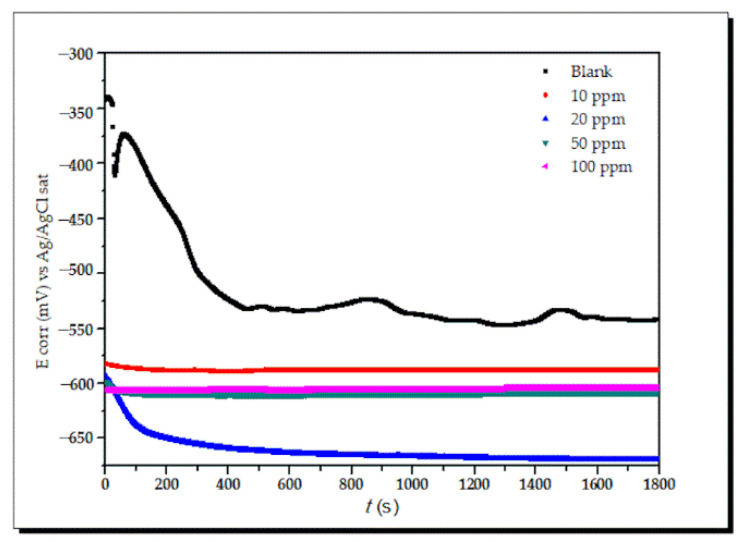
Variation of OCP at different concentrations of Lidocaine in API 5L X70 steel.

**Figure 6 ijms-23-05086-f006:**
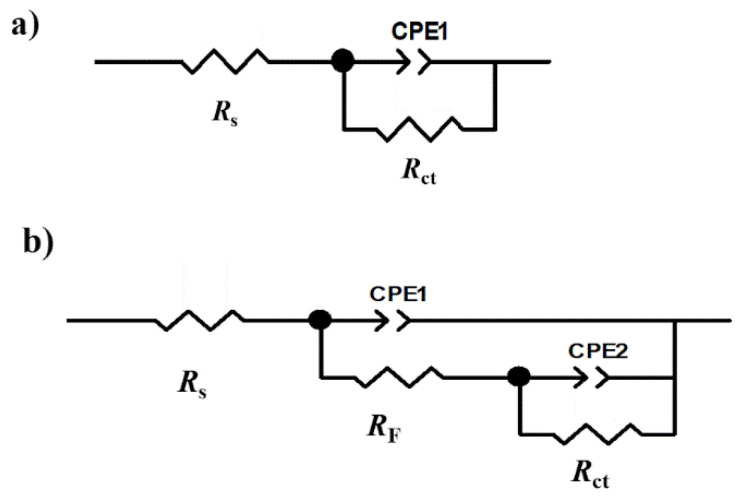
Equivalent electrical circuits. (**a**) A Randles circuit in the case of the sample was used without inhibitor (Blank). (**b**) A parallel circuit with two constant phase elements for the samples with inhibitor.

**Figure 7 ijms-23-05086-f007:**
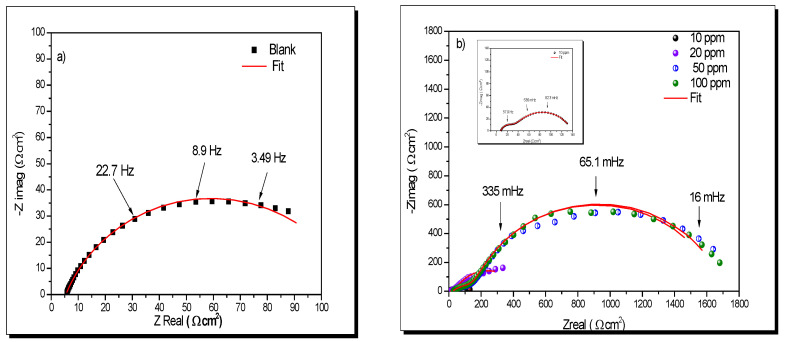
Nyquist plots (**a**) without inhibitor and (**b**) different concentrations of lidocaine in API 5L X70 immersed in NaCl 3%.

**Figure 8 ijms-23-05086-f008:**
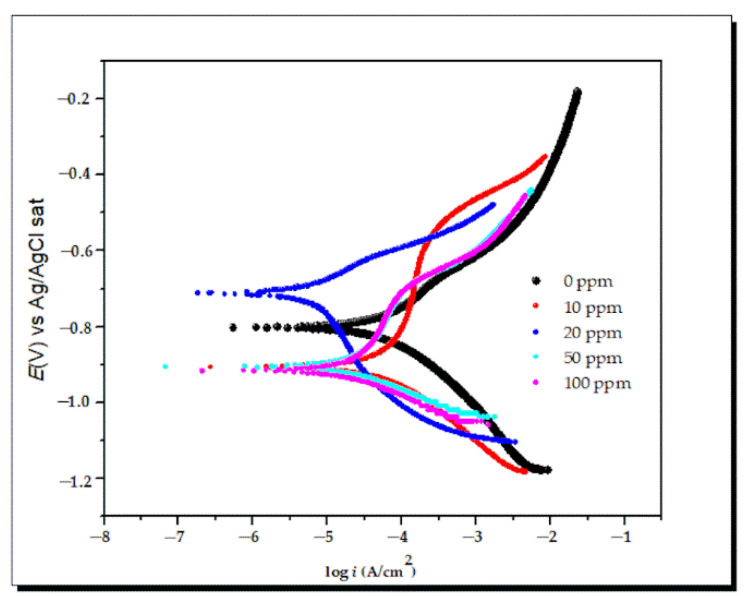
Polarization curves of different concentrations of lidocaine in API 5L X70 steel immersed in 3% NaCl.

**Figure 9 ijms-23-05086-f009:**
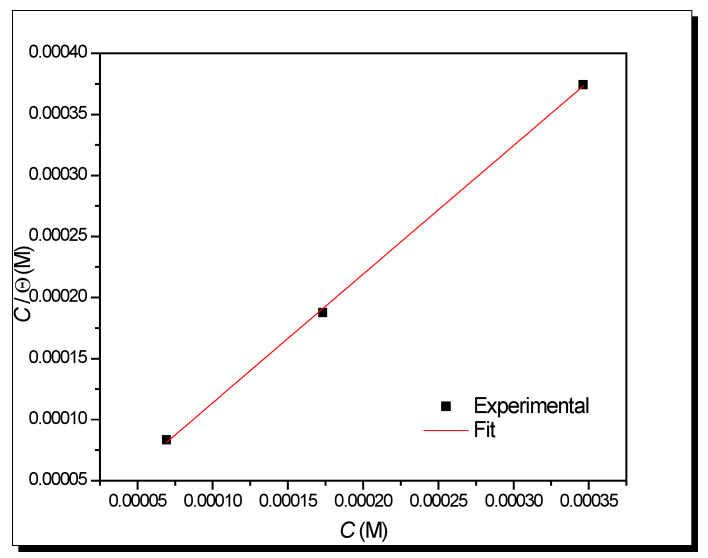
Langmuir isotherm at different concentrations of lidocaine in API 5L X70 steel immersed in NaCl 3%.

**Figure 10 ijms-23-05086-f010:**
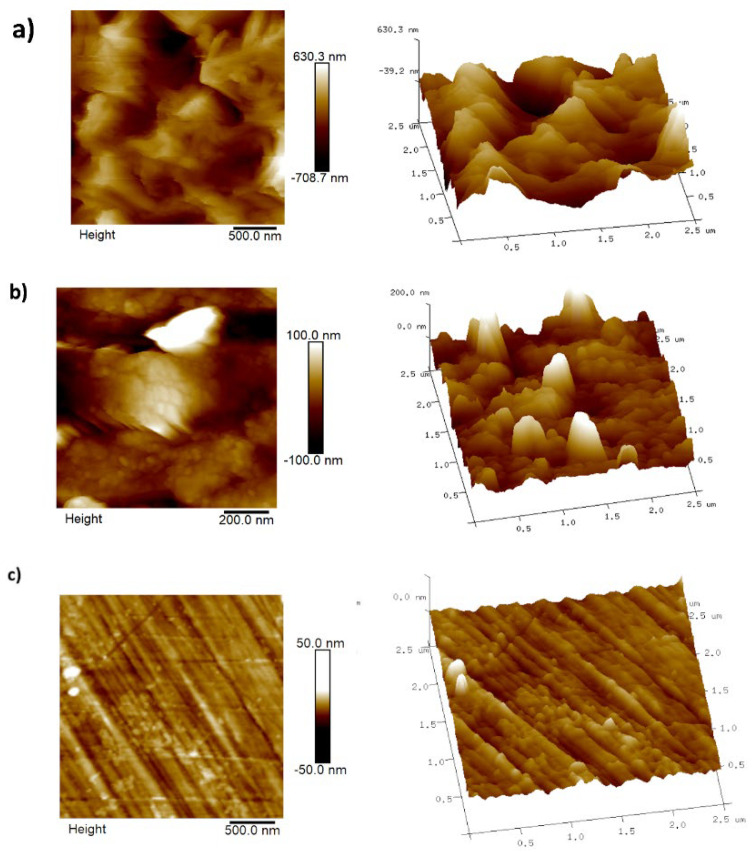
AFM images (2D and 3D formats) were recorded on the surface of API 5L X70 steel samples after 24 h immersion in NaCl 3%: (**a**) in the absence and (**b**) in the presence of 50 ppm lidocaine. The image in (**c**) corresponds to the as-polished steel sample not immersed in the corrosive media.

**Table 1 ijms-23-05086-t001:** List of variables included, in the database, used to obtain the QSAR model, symbols, units, and references of their use in QSAR studies for corrosion inhibition. Additionally, parameters and ERR were obtained by the FROLS algorithm after processing the model of Equation (5) for the final model.

*x* _i_	Descriptor	Symbol	Units	Reference	Parameter	ERR (%)
*x* _1_	Molecular weight	MW	Da	[69,70]	-	-
*x* _2_	Acid dissociation constant	p*K*_a_	-	-	0.5287	0.0600
*x* _3_	Octanol-water partition coefficient	log *P*	-	[71,72]	-	-
*x* _4_	Water solubility	log *S*	-	-	-	-
*x* _5_	Polar surface area	PSA	Å^2^	[38,71,72]	-	-
*x* _6_	Polarizability	*α*	Å^3^	[72]	-	-
*x* _7_	Energy of HOMO	*E* _HOMO_	eV	[38,69,71,72]	812.1748	97.8259
*x* _8_	Energy of LUMO	*E* _LUMO_	eV	[38,69,71,72]	823.4630	0.1034
*x* _9_	Electrophilicity	*ω*	eV	[44,72]	6579.0080	0.0688
*x* _10_	The fraction of electrons shared	Δ*N*	-	[44,70,72]	33.1669	1.3933
				**Sum of ERR**		**99.4514**

**Table 2 ijms-23-05086-t002:** Descriptor values, common use, and structure of drugs predicted as corrosion inhibitors with efficiency above 95%.

Drug	p*K*_a_	*E* _HOMO_	*E* _LUMO_	*ω*	Δ*N*	IE%	Common Use	2D Structure
Deserpidine	6.68	−4.92	−2.42	0.92	1.33	95.29	Antihypertensive and antipsychotic	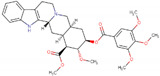
Daunorubicin	8.20	−5.87	−4.01	1.23	1.11	96.67	Cancer treatment	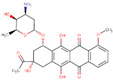
Dipyridamole	6.40	−4.34	−1.83	0.77	1.55	97.28	Anticoagulant	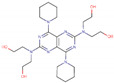
Doxorubicin	9.46	−5.86	−4.00	1.23	1.12	97.40	Cancer treatment	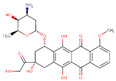
Amphotericin B	3.58	−5.27	−3.29	1.07	1.37	97.55	Antibiotic and fungicide	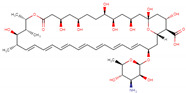
Minocycline	2.30	−5.32	−3.42	1.09	1.38	97.58	Antibiotic	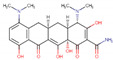
Acepromazine	9.30	−4.92	−2.53	0.93	1.37	97.73	Antipsychotic	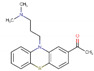
Cephaloridine	3.40	−5.31	−3.43	1.09	1.40	98.57	Antibiotic	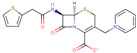
Mercaptopurine	7.80	−5.03	−2.83	0.98	1.40	98.66	Cancer treatment	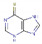
Rifampicin	1.70	−4.86	−2.81	0.96	1.55	98.71	Antibiotic	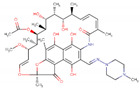

**Table 3 ijms-23-05086-t003:** Electrochemical parameters at different concentrations of lidocaine in API 5L X70 immersed in NaCl 3%.

*C* (ppm)	*R*_s_ (Ω cm^2^)	*n*	*C*_dl_ (µF/cm^2^)	*R*_ct_ (Ω cm^2^)	*C*_F_ (µF/cm^2^)	*n*2	*R*_mol_ (Ω cm^2^)	*R*_total_ (Ω cm^2^)	IE% (%)
0	6	0.800	2960	127	-	-	-	-	-
10	8.24	0.80	181.3	102.00	4034.0	0.8	28.70	130.70	3.2
20	10.53	0.77	187.5	404.10	622.2	0.52	337.90	742.00	83.0
50	24.66	0.85	90.3	1493.00	40.7	0.49	151.70	1644.70	92.3
100	24.29	0.84	51.9	1522.00	26.0	0.48	157.00	1679.00	92.5

**Table 4 ijms-23-05086-t004:** Polarization parameters for lidocaine in API 5L X70 immersed in NaCl 3%.

*C* (ppm)	*E*_corr_ (mV) vs. Ag/AgCl sat	*i_corr_* (µA/cm^2^)	*b_a_* (mV/dec)	*−b_c_* (mV/dec)	IE% (%)
*0*	−804.7	67.4	159.5	173	-
*10*	−909.7	65.0	146.6	161.5	3.4
*20*	−709.6	4.9	104.5	204.1	92.6
*50*	−907.7	7.4	170.5	60.3	89.0
*100*	−916.5	8.2	187.8	68.2	87.4

**Table 5 ijms-23-05086-t005:** Roughness values calculated from the AFM images shown in Figure 10.

*AFM* image	R_a_ (nm)	R_q_ (nm)
a	142	181
b	30.5	45
c	3.4	4.3

## Data Availability

The data presented in this study are available on request from the corresponding author.

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
