# Peer review of "A General Use QSAR-ARX Model to Predict the Corrosion Inhibition Efficiency of Drugs in Terms of Quantum Mechanical Descriptors and Experimental Comparison for Lidocaine"

_ijms, 2022, doi:10.3390/ijms23095086_

Round 1

Reviewer 1 Report

  1. The title of the manuscript needs to be more specific

2- The author should compare and justify, how these inhibitors are better than the already reported similar type of inhibitors in literatures.

3-- Qualitative informations are missing in abstract. Abstract should be concise and the authors need to improve with more specific short results.

4-the construction of the introduction is not correct because the author confused material method and introduction, he explains:

1.1. Corrosion Inhibition and QSAR Fundamentals

1.2. ARX modeling

...

The introduction should be rephrased and use recent references to compare the results you found with what is in the literature

5-The purity of the product used must be added in the solution preparation (paragraph 2.2. Experimental details)

6-You say in paragraph 2.2.1. Preparing the solution...the initial concentration of the solution was 0.01 M, dissolved in ethanol, monkeys you are using NaCl 3/ as the corrosive medium, so the product is not soluble in NaCl 3/ how can it be used as a corrosion inhibitor? Justify

7-Reduce the material method part and give the name of the results part as results and discussions

8-Add the polarization curves to confirm the results you found in the EIS curves

9-

-The EIS curve of the Blank may not be correct, justified by the addition of a reference

-In the presence of inhibitor the curve illegible separates the curves by the use of colors

-In the presence of inhibitor there is the appearance of two well separated loops the first corresponds to the inhibitor film, not the polarization resistance (Rp is the sum of the resistances) correct it

-Revise the results of table 3

10-In the Adsorption process you use in equation (14) 55.5, this equation is used if the concentration of the inhibitor is expressed in mol/L. How do you explain that?

  1. Some advance surface study such as AFM and XPS should be added.

Author Response

  1. The title of the manuscript needs to be more specific

Reply: The title was replaced by another one more specific, highlighting the property modeled, as follows:

“A general use QSAR-ARX model to predict the corrosion inhibition efficiency of drugs in terms of quantum mechanical descriptors and experimental comparison for lidocaine.”

  1. The author should compare and justify, how these inhibitors are better than the already reported similar type of inhibitors in literatures.

Reply: We thank the reviewer’s comment. In order to highlight the novelty of the corrosion inhibition efficiencies predicted in comparison with those reported in the literature, the following discussion was included in section 3.3 “high-efficiency corrosion inhibitors” of the revised version of the manuscript.

  • The above discussion points out the highly efficient corrosion inhibition properties expected for ten commercial drugs, with IE% values ranging from 95.29 to 98.71% (Table 2). In comparison, other drugs experimentally evaluated exhibited similar or lower corro-sion inhibition efficiencies. For instance, losartan, a drug commonly used for hypertension treatment, obtained a maximum IE%, by EIS technique, of only 92.0% [88]. Also, salbuta-mol, a commonly used treatment for asthma, was obtained with a maximum IE%, deter-mined by EIS measurement, of about 84% [89]. Similarly, Anadebe and coworkers collect-ed the maximum IE% achieved by several recently reported commercial drugs, ranging from 80 to 95% [89]. Drugs applied to carbon steel were tobramycin (80%) [90], metformin (90%) [91], metolazone (92%) [92] and nifedipine (94%) [93]. Also, species applied on mild steel surfaces were: dexamethasone (83%) [94], rosuvastatin (90%) [95], ambroxol (94%) [96] and dapsone (95%) [97]. Also, Abeng, et al. reported the IE% on carbon steel achieved by moxifloxacin (88.2%), nifedipine (89.6%), metolazone (92.8%), and levofloxacin (94.1%) [98]. Even more, the recently evaluated drugs, mycophenolic acid [25] and fluconazole [24] obtained moderate efficiencies of about 90%. Clearly, all these recently studied drugs ob-tained values below the threshold assumed in this work. However, other drugs were comparable to those expected for the molecules analyzed in this subsection. This is the case of pyrazinamide, isoniazid, and rifampicin, which achieved maximum IE% values of about 95.86, 97.89, and 97.06%, respectively [99]. Thus, the predictions and tendencies discussed could be the object of study and further confirmation

  1. Qualitative informations are missing in the abstract. Abstract should be concise and the authors need to improve with more specific short results.

Reply: We thank the reviewer’s suggestion. The abstract now includes qualitative as well as quantitative information. Also, to be more specific, short results as comparison metric were included in the abstract.

  • Abstract: A study of 250 commercial drugs to act as corrosion inhibitors on steel has been de-veloped by applying the quantitative structure-activity relationship (QSAR) paradigm. Hard-soft acid-base (HSAB) descriptors were used to establish a mathematical model to predict the corrosion inhibition efficiency (IE%) of several commercial drugs on steel surfaces. These descriptors were calculated through third-order density-functional tight binding (DFTB) methods. The mathemat-ical modeling was carried out through autoregressive with exogenous inputs (ARX) framework and tested by fivefold cross-validation. Another set of drugs was used as an external validation, obtaining SD, RMSE and MSE, obtaining 6.76, 3.89, 7.03 and 49.47 %, respectively. With a predicted value of IE% = 87.51%, lidocaine was selected to perform a final comparison with experimental results. By the first time, this drug obtained a maximum IE%, determined experimentally by electrochemical impedance spectroscopy measurements at 100 ppm concentration, of about 92.5%, which stands within limits of 1 SD from the predicted ARX model value. From the qualitative perspective, several potential trends have emerged from the estimated values. Among them, macrolides, alkaloids from Rauwolfia species, cephalosporin, and rifamycin antibiotics are expected to exhibit high IE% on steel surfaces. Also, IE% increases as the energy of HOMO decreases. The highest efficiency is obtained in case of the molecules with the highest ω and ΔN values. The most efficient drugs are found with pKa ranging from 1.70 to 9.46. The drugs recurrently exhibit aromatic rings, carbonyl, and hydroxyl groups with the highest IE% values.
  1. The construction of the introduction is not correct because the author confused material method and introduction, he explains:

1.1. Corrosion Inhibition and QSAR Fundamentals

1.2. ARX modeling.

...

Reply: The structure and numbering of the introduction was revised, and a few topics were relocated. Now, NARMAX and ARX subsections appear in consecutive order in theoretical methods. Subsections were renumbered as follows:

  1. Introduction
    • Corrosion inhibition and QSAR fundamentals
    • QSAR paradigm and HSAB descriptors
  2. Theoretical and experimental methods
    • NARMAX system identification approach
    • ARX theoretical model
      • Arrangement of candidate terms
      • FROLS and ERR algorithms for model structure selection
      • Cross-Validation
    • Experimental details
      • Solution preparation
      • Electrochemical evaluation
  1. Results and discussion
    • Model determination
      • Data processing into an ARX linear system
      • Term selection through FROLS and ERR
    • Main tendencies
    • High-efficiency corrosion inhibitors
    • Experimental verification
  2. Conclusions

The introduction should be rephrased and use recent references to compare the results you found with what is in the literature.

Reply We have corrected the introduction, reducing it by 36 % from the previous version. Also, several recent references were included, in the revised version, to compare the ARX model with those found in the literature. Also, other models reporting common metrics discussed.

5-The purity of the product used must be added in the solution preparation (paragraph 2.2. Experimental details) 

            Reply: In the final version we added this information.

  • ……. Different concentrations of the Lidocaine compound (figure 1) were prepared obtained in injectable solution from Ahorro pharmacies with 98 % purity……

6-You say in paragraph 2.2.1. Preparing the solution...the initial concentration of the solution was 0.01 M, dissolved in ethanol, monkeys you are using NaCl 3/ as the corrosive medium, so the product is not soluble in NaCl 3/ how can it be used as a corrosion inhibitor? Justify 

Reply: In this case, the starting point is a mother solution with the indicated concentration. to later make the dilutions

7-Reduce the material method part and give the name of the results part as results and discussions

Reply: Section 3 was renamed as Results and discussion. Material methods was reduced from the previous version; however, a small part of the introduction was moved to that part, leading to one hundred additional words.

8-Add the polarization curves to confirm the results you found in the EIS curves 

            Reply: we added the polarization curves as follows:

  • ……………..Electrochemical evaluation

  • The standard three-electrode system was used for the electrochemical evaluation at room temperature. The API 5L X70 sample was the working electrode, a saturated Ag/AgCl electrode as the reference electrode, and a graphite bar as the counter electrode. The test sequence was performed on a piece of Gill-AC equipment as follows: a) Open cir-cuit potential (OCP) was measured for 1800 seconds, b) Electrochemical impedance spec-troscopy (EIS) was employed using 10–2 – 104 Hz with an amplitude of ±10 mV. The ex-posure area of experimentally used samples was 0.78 cm2. The electrochemical tests were performed in triplicate.
  • After the EIS measurements, potentiodynamic polarization curves of the inhibitor at different concentrations were performed, measured from –300 mV to 300 mV in relation to the open circuit potential (OCP), with the speed of 60 mV/min using the ACM Analysis software for data interpretation.

  • Polarization curves

  • The polarization curves of the API 5L X70 steel, immerse in 3% NaCl with and without lidocaine, are shown in figure 8. The polarization parameters are enlisted in table 4: corrosion potential (Ecorr), current density (icorr), tafel slopes (ba and bc), and the inhibition efficiency (IE%) was determined by:
  • IE% = [1 – icorr / icorr blank] × 100                                                              (14)
  • Where icorr and icorr blank are the current density with and without inhibitor.

  • Figure 8 shows that the curve shifts to the left due to the corrosion current density decreasing when the concentration increases due to the adsorption of the inhibitor.

  •  
  • Figure 8. Polarization curves of different concentrations of lidocaine in API 5L X70 steel immerse in 3% NaCl
  • The electrochemical parameters by this technique show that the corrosion current density (icorr) value decreased in the presence of the lidocaine inhibitor, being attributed to the protected metal surface (Table 4). This phenomenon implies that the inhibitor might suppress the anodic reaction of the metal dissolution and the detachment of cathodic hydrogen[105]. While the inhibition efficiency by this technique shows similar results to the other technique (EIS), and the best concentration was 20 ppm with 92.6 % to protect the metal surface. Finally, according to corrosion potential (Ecorr), at a concentration of 20 ppm, the lidocaine behaved as an anodic inhibitor, while at 10, 50, and 100 ppm, the behavior was cathodic.

C

(ppm)

Ecorr

(mV) vs Ag/AgCl sat

icorr

(µA/cm2)

ba

(mV/dec)

-bc

(mV/dec)

IE%

(%)

0

–804.7

67.4

159.5

173

-

10

–909.7

65.0

146.6

161.5

3.4

20

–709.6

4.9

104.5

204.1

92.6

50

–907.7

7.4

170.5

60.3

89.0

100

–916.5

8.2

187.8

68.2

87.4

Table 4. Polarization parameters for Lidocaine in API 5L X70 immersed in NaCl 3%

9- The EIS curve of the Blank may not be correct, justified by the addition of a reference 

-In the presence of inhibitor the curve illegible separates the curves by the use of colors 

-In the presence of inhibitor there is the appearance of two well separated loops the first corresponds to the inhibitor film, not the polarization resistance (Rp is the sum of the resistances) correct it 

            Reply: We changed the Nyquist plot as follows:

On the other hand, the Nyquist diagram in Figure 7b in all concentrations showed two processes: one attributed to inhibitor film and the charge transfer resistance [103].

Figure 7. Nyquist plots a) without inhibitor and b) different concentrations of lidocaine in API 5L X70 immersed in NaCl 3%

-Revise the results of table 3 

                        Reply: We revise the table 3 as follows:

Table 3. Electrochemical parameters at different concentrations of lidocaine in API 5L X70 immersed in NaCl 3%.

C

(ppm)

Rs

(Ω cm2)

n

Cdl

(µF/cm2)

Rct

(Ω cm2)

CF

(µF/cm2)

n2

Rmol

(Ω cm2)

Rtotal

(Ω cm2)

IE%

(%)

0

6

0.800

2960

127

-

-

-

-

-

10

8.24

0.80

181.3

102.00

4034.0

0.8

28.70

130.70

3.2

20

10.53

0.77

187.5

404.10

622.2

0.52

337.90

742.00

83.0

50

24.66

0.85

90.3

1493.00

40.7

0.49

151.70

1644.70

92.3

100

24.29

0.84

51.9

1522.00

26.0

0.48

157.00

1679.00

92.5

10-In the Adsorption process you use in equation (14) 55.5, this equation is used if the concentration of the inhibitor is expressed in mol/L. How do you explain that? 

Reply:  The conversion from ppm to mol /L was made

  1. Some advance surface study such as AFM and XPS should be added. 

            Reply: We added the results of AFM as follows:

  • 4. Experimental verification

…. e) AFM analysis

 Figure 10 shows AFM images recorded on the surfaces of steel samples, and Table 5 shows the roughness values, both Ra (the mean roughness or arithmetic average of the absolute values of the roughness profile ordinates) and Rq (root mean square roughness or the root mean square average of the roughness profile ordinates) is reported. After exposure to the corrosive media for 24 h in the absence (Figure 10a) and in the presence of 50 ppm of lidocaine (Figure 10b), it can be noted that the roughness values of the steel sample that was not protected with lidocaine are notorious compared with that of the sample that was protected with lidocaine or that which was not exposed to the corrosive media, Figure 10c, such notable rugosity is due to the different corrosion products formed in each case.  

Figure 10. AFM images (2D and 3D formats) were recorded on the surface of API 5L X70 steel samples after 24 h immersion in NaCl 3%; a) in the absence and b) in the presence of 50 ppm lidocaine. The image in c) corresponds to the as-polished steel sample not immersed in the corrosive media.

 Table 5. Roughness values calculated from the AFM images shown in Figure 9.

AFM

image

Ra

(nm)

Rq

(nm)

a

142

181

b

30.5

45

c

3.4

4.3

Reviewer 2 Report

The author reported on a study titled: “QSAR Study of Common Drugs Towards their Reuse as Corrosion Inhibitors on Steel and Experimental Comparison for Lidocaine”. The perusal of the literature indicates that this work has advanced the use of drugs as corrosion inhibitors by adopting artificial intelligence in the selection of new druglike molecules for corrosion protection.  There are minor revisions to be carried out before publication of the manuscript.

Specific Comments

  1. Define using equations the criteria for the model performances such as MAPA and SD.
  2. Define the meaning of all parameters in Table 1.
  3. The English language needs more improvement.

Author Response

Comments and Suggestions for Authors

The author reported on a study titled: “QSAR Study of Common Drugs Towards their Reuse as Corrosion Inhibitors on Steel and Experimental Comparison for Lidocaine”. The perusal of the literature indicates that this work has advanced the use of drugs as corrosion inhibitors by adopting artificial intelligence in the selection of new druglike molecules for corrosion protection.  There are minor revisions to be carried out before publication of the manuscript.

Specific Comments

  1. Define using equations the criteria for the model performances such as MAPA and SD.

Reply: Mean absolute percentage error, standard deviation, root-mean-square-error and mean square error are now defined and discussed in the manuscript as follows:

  • Other metrics were employed to analyze the model’s performance: Mean absolute percentage error (MAPE), standard deviation (SD), mean square error, and root mean square error (RMSE). These metrics, discussed throughout the manuscript, are defined now: MAPE = (1/n) Σni=1|(yi yÌ‚i)/yi|×100%, SD = [(1/n – 1)Σni=1 (yÌ… yi)2]1/2, MSE = (1/n) Σni=1 ( yiyÌ‚i )2 and RMSE = [(1/n) Σni=1 ( yiyÌ‚i )2 ]1/2. Where yi is the experimental value of IE% for compound i, Å·i is the estimated value of IE% for compound i provided by the model, and yÌ… is the average IE% within a sample of compounds of size n.

  • The testing set included aspirin, cephapirin, ascorbic acid, imidazole, trimethoprim, clindamycin, phenobarbital, and doxycycline data, obtaining MAPE, SD, RMSE and MSE of about 5.18, 2.51, 4.87 and 23.80 %, respectively.

  • MAPE, SD, RMSE and MSE, obtaining 76, 3.89, 7.03 and 49.47 %, respectively.

  • Finally, four different experiments, pipelines, were done by the extra trees regressor algorithm. The model obtained improved the external comparison only to 5.44, 2.91, 5.35, 28.59 % for MAPE, SD, RMSE and MSE, respectively

  1. Define the meaning of all parameters in Table 1.

Reply: We take into consideration the reviewer’s comment. The following description of all the parameters in table 1 was included.

Where y is an output term plus ten input terms composed of candidate descriptors, as labeled in Table 1, and a corresponding parameter value or weight βi. The list of the ten candidate descriptors and their corresponding index number is listed in Table 1. Each drug's molecular weight (MW) was considered a size-dependent parameter. Also, the negative base-10 logarithm of the acid dissociation constant of a solution, pKa = – logKa, was included to determine the strength of an acid in the solution. Also, the octanol-water partition coefficient, log P, is a descriptor associated with the concentration of a given substance in the aqueous phase of a two-phase octanol-water mix. Similarly, the log S descriptor is directly related to the water solubility of a substance employing a base-10 logarithm. Besides, the polar surface area (PSA) is the molecular surface associated with heteroatoms and polar hydrogen atoms, giving a quantitative amount related to charge accumulation. In addition, polarizability, α, denotes the tendency of a particular molecule to acquire an electric dipole moment in the presence of an external electric field. As described previously, energies of HOMO and LUMO orbitals can be related, through Koopman's theorem, to ionization energy and the electron affinity of a given molecular species, respectively. In addition, electrophilicity, ω, relates to the change in energy of an electrophile when it comes in contact with a perfect nucleophile, being a measure of the tendency to react between electrophile and nucleophile species. Finally, the fraction of electrons shared, ΔN, was selected since it relates to the amount of charge transferred from one species to another (Table 1).

  1. The English language needs more improvement.

Reply: We thank the reviewer’s suggestion. The current version of the manuscript was carefully revised

Reviewer 3 Report

Manuscript ID: ijms-1665507 entitled:

QSAR study of common drugs towards their reuse as corrosion inhibitors on steel and experimental comparison for lidocaine.

Authors Carlos Beltrán-Perez , Andrés A.A. Serrano , Gilberto Solis-Rosas , Araceli Espinoza-Vazquez * , Alan Miralrio *

General comment

The paper presents a study regarding 250 commercial drugs as candidates to act as corrosion inhibitors on steel. The Quantitative Structure-Activity Relationship (QSAR) paradigm was used.

Some recommendations and observation are listed below:

  1. In the Experimental section it is useful to specify the purity of the compound lidocaine (lidocaine was used as it was received from the pharmacy "Farmacias del Ahorro", can it contain other compounds?). Mention in protocol for EIS the working potential, temperature, if from solution oxygen was removed. Number of replicates.

  1. Make a legend with abbreviation of each column in Table presented in supplementary data and specify in the main text.

  1. Some value presented in text or in Table 2, is not correlated with those presented in table in supplementary data:

Four drugs excluded from the analyses due to their unrealistic IE% values, present in text a different value than in supplementary data: methacycline (105.68%), glycine (128.90%), ethosuximide (168.04%) and hexetidine (280.91%). For example Glycine IE% in text 128.90% and 124.3% in table, Hexitidine IE% in text 280.91% and 259.25% in table.

For sulfapyridine the error in the IE prediction exceeds the presented value.

  1. Use the same notation for all resistance and CPE elements in EEC (Fig 6), text and eq 9.

In Table 3, specify the meaning of CF, Rct, Rmol, Rtotal to avoid any confusion. , add the incertitude for each parameter. Specify for clarity in the text the equivalent electric circuits (EEC) used for blank and EEC used in the case of inhibitor.

  1. Authors at R 570 write that Table 3 shows the electrochemical parameters obtained after fitting with the equivalent electric circuits of Figure 5, but Fig 5 represent the OCP. Correct in the text.

  1. Check the values in table 3. The eq 9 give slightly different values for IE.

Simplify and correct the eq 9 finally will give IE = (Rctinh-Rctblank)/Rct inh.

  1. Precise (eq 13) the calculation mode of the surface coverage, ÆŸ (gravimetric or from EIS data).

  1. This paper needs a uniform formatting style. (ie R589-R590) and a description of the units of measurement for all parameters in the inserted equations.
  2. A nonlinear model for the study of corrosion interaction of inhibitors with metal surface in saline solutions can be take in account and could give better results?
  3. From the analysis of the properties presented in the additional data, it appears that the analysis of the molecular structure of corrosion inhibitors apparently is not sufficient to provide an adequate explanation of the inhibition performance determined experimentally. The computational investigation is able to restrict research to certain families of compounds. The intertwining of various effects such as molecular structure, solubility, pH, double layer effects, temperature, etc alter the result. Also in discussion, other factors such as cost, toxicity, availability and environmental friendliness must be taken into account. Can the author comment regarding these aspects?

Author Response

QSAR study of common drugs towards their reuse as corrosion inhibitors on steel and experimental comparison for lidocaine.

Authors Carlos Beltrán-Perez , Andrés A.A. Serrano , Gilberto Solis-Rosas , Araceli Espinoza-Vazquez * , Alan Miralrio *

General comment

The paper presents a study regarding 250 commercial drugs as candidates to act as corrosion inhibitors on steel. The Quantitative Structure-Activity Relationship (QSAR) paradigm was used.

Some recommendations and observation are listed below:

  1.  In the Experimental section it is useful to specify the purity of the compound lidocaine (lidocaine was used as it was received from the pharmacy "Farmacias del Ahorro", can it contain other compounds?). Mention in protocol for EIS the working potential, temperature, if from solution oxygen was removed. Number of replicates.

Reply: The electrochemical evaluation wasn´t removed oxygen and the working potential was an open circuit potential. On the other hand, we added this information in the final version as follows

  • ……… Electrochemical evaluation

The standard three-electrode system was used for the electrochemical evaluation at room temperature. The API 5L X70 sample was the working electrode, a saturated Ag/AgCl electrode as the reference electrode, and a graphite bar as the counter electrode. The test sequence was performed on a piece of Gill-AC equipment as follows: a) Open circuit potential (OCP) was measured for 1800 seconds, b) Electrochemical impedance spectroscopy (EIS) was employed using 10–2 – 104 Hz with an amplitude of ±10 mV. The exposure area of experimentally used samples was 0.78 cm2. The electrochemical tests were performed in triplicate.

  1. Make a legend with abbreviation of each column in Table presented in supplementary data and specify in the main text.

Reply: The following statement about the data contained in the database is now included in supplementary materials section.

Supplementary Materials: The entire database built for the construction of the ARX model can be downloaded (LINK HERE). The rows in the database are ordered as follows: Common name, International Union of Pure and Applied Chemistry (IUPAC) name, Chemical Abstracts Service (CAS) number, chemical formula, number of C atoms, number of H atoms, number of N atoms, number of O atoms, number of S atoms, number of P atoms, canonical Simplified molecular-input line-entry system (SMILES), IUPAC International Chemical Identifier (InChl), molecular weight (MW), acid dissociation constant at logarithmic scale pKa, octanol-water partition coefficient logP, water solubility coefficient logS, polar surface area (PSA), polarizability (α), hydrogen acceptor count, hydrogen donor count, energy of the highest occupied molecular orbital (HOMO) -1, energy of HOMO, energy of the lowest unoccupied molecular orbital (LUMO), ionization energy, electron affinity, electronegativity, global hardness, electrophilicity (ω), fraction of electrons transferred (ΔN), experimental corrosion inhibition efficiency (IE%), ARX and Watson predictions.

  1. Use the same notation for all resistance and CPE elements in EEC (Fig 6), text and eq 9. In Table 3, specify the meaning of CF, Rct, Rmol, Rtotal to avoid any confusion, add the incertitude for each parameter. Specify for clarity in the text the equivalent electric circuits (EEC) used for blank and EEC used in the case of inhibitor.

            Reply: we added the observation as follows:

  • The equivalent electric circuits employed to fit the experimental data of the Nyquist diagrams are shown in Figure 6. A Randles circuit in the case of the sample of Figure 1a was used without inhibitor (Blank). A parallel circuit with two constant phase elements (Figure 1b) for the samples with inhibitor.

  •  
  •  
  • Figure 1. Equivalent electrical circuits

  • Where Rs is the solution resistance, Rct is the charge transfer resistance, CPEinh is the constant phase element of the inhibitor, and CPERct is the constant phase element associated with the double layer. The inhibitor efficiency IE% was calculated by the equation [100]:
  • IE% = 100 [(Rp blank–1Rp inh–1)/ Rp blank–1)] (9)
  • Where Rp blank–1 is the polarization resistance of blank and Rp in–1 is the polarization resistance of sample with inhibitor.

  • The polarization resistance (Rp) was calculated with:

  • Rp = Rct + RF                                                    (10)
  • Where Rct is charge transference resistance and RF film resistance, in Ω cm2.

  1. Authors at R 570 write that Table 3 shows the electrochemical parameters obtained after fitting with the equivalent electric circuits of Figure 5, but Fig 5 represent the OCP. Correct in the text.

Reply: we agree with this observation

  • …….. Table 3 shows the electrochemical parameters obtained after fitting with the equivalent electric circuits of Figure 6………..

  1. Check the values in table 3. The eq 9 give slightly different values for IE.

Simplify and correct the eq 9 finally will give IE = (Rctinh-Rctblank)/Rct inh.

Reply: We changed the values inhibition as follows

  • Table 3. Electrochemical parameters at different concentrations of lidocaine in API 5L X70 immersed in NaCl 3%.

C

(ppm)

Rs

(Ω cm2)

n

Cdl

(µF/cm2)

Rct

(Ω cm2)

CF

(µF/cm2)

n2

Rmol

(Ω cm2)

Rtotal

(Ω cm2)

IE%

(%)

0

6

0.800

2960

127

-

-

-

-

-

10

8.24

0.80

181.3

102.00

4034.0

0.8

28.70

130.70

3.2

20

10.53

0.77

187.5

404.10

622.2

0.52

337.90

742.00

83.0

50

24.66

0.85

90.3

1493.00

40.7

0.49

151.70

1644.70

92.3

100

24.29

0.84

51.9

1522.00

26.0

0.48

157.00

1679.00

92.5

  1. Precise (eq 13) the calculation mode of the surface coverage, ÆŸ (gravimetric or from EIS data).

Reply: We added this observation as follows:

  • The corrosion inhibition displaces the water molecules and replaces them with the inhibitor molecules on the metal surface.
  • Nevertheless, the superficial coverage (θ) for the different lidocaine concentrations as corrosion inhibitors in this system was evaluated by EIS using IE%:

  • θ = ( 1 / 100 ) IE%                                                                                (15)

  • Using the Langmuir isotherm, a good fit is obtained, and according to the value of the free energy of adsorption of Gibbs (equation 17), the combined process continues [106].

  • C / θ = kads–1 + C (16)
  • ΔG0ads = – 55.5 RT ln kads (17)

  1. Some value presented in text or in Table 2, is not correlated with those presented in table in supplementary data: Four drugs excluded from the analyses due to their unrealistic IE% values, present in text a different value than in supplementary data: methacycline (105.68%), glycine (128.90%), ethosuximide (168.04%) and hexetidine (280.91%). For example Glycine IE% in text 128.90% and 124.3% in table, Hexitidine IE% in text 280.91% and 259.25% in table.

Reply: We thank the reviewer’s observation. Consequently, all IE% values, extracted from supporting information, were revised and corrected. All revised values are highlighted in the manuscript. The most notable case is minocycline, with a corrected value of about 97.58%. Performance metrics, of the original version, were preserved since all calculations were based on the database. 

  • These species are sulfadiazine (106.31%), methacycline (111.72%), glycine (124.03%), ethosuximide (158.88%), and hexetidine (259.25%).

  1. For sulfapyridine the error in the IE prediction exceeds the presented value.

Reply: The metrics used to evaluate the error refers to the mean absolute error. Thus, larger errors than MAPE, of about 5.18 %, can be observed.

  1. This paper needs a uniform formatting style. (ie R589-R590) and a description of the units of measurement for all parameters in the inserted equations.

Reply: Units and expresions, of all the quantities through the manuscript, were reformated in a uniform style. Also, the following paragraphs were edited as follows:

  • ………. The polarization resistance (Rp) was calculated with:

  • Rp = Rct + RF                                                    (10)
  • Where Rct is charge transference resistance and RF film resistance, in Ω cm2.

  • The electrochemical double-layer capacitance (Cdl) was calculated through the next equation [101]:
  • Cdl = Y01/n (Rs–1 + Rct–1) (n – 1) / n (11)
    • Where Y0 is the constant phase element, Rs is the solution resistance (Ω cm2), and Rct is the charge transfer resistance (Ω cm2).
    • For the description of a frequency-independent phase shift between an applied AC potential and its current response, a constant phase element (CPE) is used, defined in the impedance representation as:

  1. A nonlinear model for the study of corrosion interaction of inhibitors with metal surface in saline solutions can be take in account and could give better results?

Reply: In order to highlight the perspectives of a high-order model to predict IE%, the following paragraph is included in the revised version of the manuscript.

  • Finally, although linear models to predict IE% are the most common approach [78], nonlinear formulations can also be suitable. A nonlinear version of ARX comes to be the NARMAX model in the current case. According to Gu, et al., nonlinear models performed better than linear ones in a study about cortical responses. Nevertheless, the linear terms had larger weights than those in the resulting NARMAX models [81]. In addition, NARMAX approaches are known to identify mathematical models for nonlinear systems, which prevail in nature. This is the case of the solar wind coupling analysis reported by Boynton and coworkers [82] or the peak air pollution levels forecasted by Pisoni, et al. [83]. Thus, it is expected that a NARMAX approach to the corrosion inhibition problem may well determine whether the phenomenon is nonlinear in terms of the proposed de-scriptors. However, as suggested by Boynton, a high number of possible monomials re-sulting from the polynomial expansion can be a challenging situation. The above stems from the need for a final parsimonious NARMAX model with fewer monomials selected out of a vast majority with no or minimal influence on the phenomenon. In principle, such as in the ARX approach used here, the FROLS algorithm could lead to a small set of mo-nomials within the selected allowable model order [84].

Reply: We agree with the reviewer’s comment. Consequently, we included a brief statement about the considerations of the model.

  • A brief discussion is included below to identify the extent and perspectives of the ARX linear model. In principle, corrosion inhibition is a multifactorial phenomenon since drug solubility, pH, temperature, concentration, corrosive medium, dynamic con-ditions, the employed alloy, and even experimental technique used to determine IE% could influence the results [2]. Thus, it is possible to assume that a 5-parameters model like that introduced above is not enough to catch the variety of conditions experimentally used. However, to the best of our knowledge, these experimental variables are not recur-rently considered, possibly by the scare IE% values measured with the same experi-mental design, hindering the formulation of mathematical models. Nevertheless, work conditions are naturally occurring variables that should be considered for robust predic-tive models.

Round 2

Reviewer 1 Report

No comment

Reviewer 3 Report

The question of the purity of lidocaine refers to the fact that in the text the authors mentioned that lidocaine was used as received from the pharmacy "Farmacias del Ahorro", but the solution for injection may contain other excipients such as 1N hydrochloric acid, sodium hydroxide. 1N (maximum allowed is 1%) and water. Perhaps a clarification would be helpful.